# Seeing the Whole in the Parts in Self-Supervised Representation Learning

## Abstract

Recent successes in self-supervised learning (SSL) model spatial co-occurrences of visual features either by masking portions of an image or by aggressively cropping it. Here, we propose a new way to model spatial co-occurrences by aligning local representations (before pooling) with a global image representation. We present CO-SSL, a family of instance discrimination methods and show that it outperforms previous methods on several datasets, including ImageNet-1K where it achieves $71.5\%$ of Top-1 accuracy with 100 pre-training epochs. CO-SSL is also more robust to noise corruption, internal corruption, small adversarial attacks, and large training crop sizes. Our analysis further indicates that CO-SSL learns highly redundant local representations, which offers an explanation for its robustness. Overall, our work suggests that aligning local and global representations may be a powerful principle of unsupervised category learning.

## 1 Introduction

Recent self-supervised learning (SSL) approaches learn visual representations that perform well on diverse downstream tasks, including object categorization. These methods include instance discrimination (ID) (Chen et al., 2020a) and masked modeling (MM) (He et al., 2022; Bao et al., 2021) approaches. While ID methods train representations to be invariant over different small crops of augmented images, MM uses a partially masked image to predict the masked information. Existing analyses suggest that the best configurations involve discarding most of the information from the images (Tian et al., 2021; He et al., 2022; Assran et al., 2023). Thus, in both cases, the learning mechanism hinges on the same learning principle, *i.e.* modeling the co-occurrences of visual features in different parts of an image. For instance, if there is an elephant trunk, the method may learn that there is likely also an elephant tusk in the image, and vice versa. Intuitively, this makes category recognition (an elephant) less sensitive to the exact visual features (trunk or tusk) present in an image.

Spatial statistical learning is also a fundamental aspect of biological vision. A classic study investigated the ability of humans to extract spatial co-occurrences among features (Fiser & Aslin, 2001). They used a set of 12 shapes, which, unknown to subjects, were organized into 6 pairs. During a familiarization phase, subjects were exposed to arrays of shapes where the members of a pair were always adjacent, without specific instructions. During a test phase, subjects had to name the more familiar pair among a previously seen pair and a new pair. Subjects were able to identify the familiar pair, confirming their ability to learn spatial regularities during familiarization. Follow-up works showed that the extraction of spatial co-occurrences is automatic (Turk-Browne et al., 2005; Fiser & Aslin, 2002) and allows the creation of perceptual "chunks" (Fiser & Aslin, 2005).

Here, we introduce a new self-supervised learning model that constructs similar visual representations for frequently co-occurring local visual features. Concretely, we propose applying an SSL loss function between local representations (right before a final pooling stage) and the global image representation at the final layer, resulting in a new family of SSL methods we call Co-Occurrence SSL (CO-SSL). Intuitively, CO-SSL "pushes" a global image representation (elephant) to lie at the center of and "attract" its co-occurring local representations (trunk, tusk, etc.). We derive two members of this family, namely CO-BYOL and CO-DINO. To fully harness CO-SSL's ability to relate representations at different spatial scales, we introduce the RF-ResNet family of neural architectures, a variation of ResNets (He et al., 2016) that extracts an averaged bag of patch representations

with small RFs after the last pooling layer. This enhances CO-SSL's ability to infer the statistical relationships between different portions of an image.

Our experiments show that CO-SSL achieves 71.5% accuracy with 100 pre-training epochs on ImageNet-1K. CO-SSL also outperforms its SSL counterparts on Tiny-ImageNet, ImageNet-100 and ImageNet-1K (10% images) and is more robust to noise corruption, internal corruption, small adversarial attacks. Our analysis shows that CO-SSL extracts more redundant local representations, offering an explanation for its robustness. In sum, our contributions are the following:

- We propose CO-SSL, a new family of SSL methods that model co-occurrences among local and global representations;
- we introduce a new CNN architecture, RF-ResNet, that extracts a bag of local patch representations to harness CO-SSL's ability to align local and global representations;
- we identify which properties of CO-SSL make it better at category recognition and more robust to diverse corruptions and small adversarial attacks.

Overall, our work demonstrates the benefits of aligning local and global image representations for seeing the whole in its parts.

## 2 RELATED WORKS

**Self-supervised ID learning.** ID methods learn to align visual representations of different views of an image, each view passing through two different data-augmentation pipelines (Chen et al., 2020a; Grill et al., 2020; Bardes et al., 2022a; He et al., 2020). Importantly, augmentations drastically alter images without perturbing their semantic content, e.g. cropping, color jittering . . . ID methods split into four approaches that mainly differ with respect to how they keep all representations different from each other: 1- contrastive learning methods explicitly make an image representation dissimilar from all other images' representations (Chen et al., 2020a; He et al., 2020; Wang & Qi, 2022; Hu et al., 2021; Wang et al., 2023b; Dwibedi et al., 2021); 2- Distillation methods use asymmetric neural network architectures to align two image representations (Grill et al., 2020; Chen & He, 2021; Gidaris et al., 2020; Caron et al., 2021); 3- Feature decorrelation methods reduce the feature redundancy within/across views' representations (Zbontar et al., 2021; Bardes et al., 2022a; Wang et al., 2023a); 4- clustering-based methods apply one of the previously mentioned method, but on clusters of image representations (Caron et al., 2020; Pang et al., 2022; Amrani et al., 2022; Estepa et al., 2023). For more information, we refer to a recent review of ID (Giakoumoglou & Stathaki, 2024). Unlike us, these works do not focus on learning similar representations for co-occurring visual features, beyond learning invariance to crops at each training iteration.

**Patch representations.** Learning patch representations in a neural network can significantly boost the supervised performance of diverse neural architectures (Trockman & Kolter, 2023; Liu et al., 2022a; Khan et al., 2022). But these works mostly focus on patch embeddings with tiny RFs (e.g. 0.5% of the image) and few non-linear layers, making difficult to extract semantic features. (Brendel & Bethge, 2018) found that simply aggregating visually local category predictions constructed with supervision works well; here, we are concerned about SSL. Recent works tend to argue that a rapid increase of the RF size through layers leads to better performance (Ding et al., 2022; Liu et al., 2022c). Here, we take an opposite stand and show that CO-SSL with small RFs can outperform standard methods. Other works employ dense SSL to learn patch representations (Bardes et al., 2022b; Wang et al., 2021; Xie et al., 2021b; Yun et al., 2022; Ziegler & Asano, 2022), but have not demonstrated strong benefits for downstream category recognition

**Spatial statistical learning.** Few SSL approaches leverage patch representations for categorization. Seminal methods solve Jigsaw puzzles by learning patch representations (Noroozi & Favaro, 2016) or pre-train representations by predicting a bag of pre-trained visual "words" (Gidaris et al., 2020). Recent ID methods heavily use the multicrop strategy (mc) (Caron et al., 2020; 2021; Hu et al., 2021; Wang et al., 2023a). ID-mc methods construct several large (typically 2) and tiny crops of an image (typically 4-8) and make the representation invariant between the crops. This improves the sample efficiency of ID methods (Caron et al., 2020; 2021). A notable adaptation of VICReg (Bardes et al., 2022a) proposes an extreme form of multi-crop with tens of crops. Once aggregated,

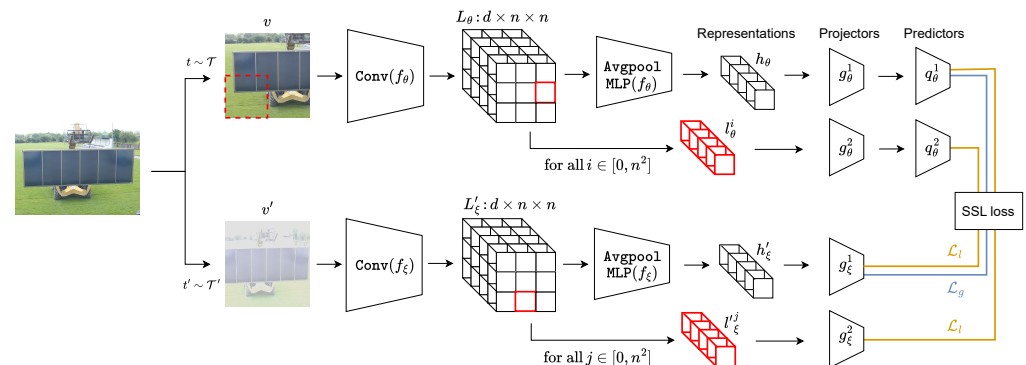

Figure 1: Architecture of CO-BYOL. We augment an image through two augmentations pipelines and forward one to the visual encoder $f_\theta$ and one to the momentum encoder $f_\xi$. Both output a set of local representations. Then, we spatially average these local representations into two global representation, which are fed into MLPs computing projections and predictions for the standard loss function $\mathcal{L}_g$ of BYOL. In CO-BYOL, we also individually compute local embeddings of an image with two new local projectors and a new local predictor. Finally we symmetrically compute the averaged BYOL loss $\mathcal{L}_l$ between each local embedding and the global embedding of the other image.

these patch representations support a good downstream categorization (Chen et al., 2022), especially with very few epochs (Tong et al., 2023). In the same line of work, DetCo (Xie et al., 2021a) applies contrastive learning between a group of local patches and the global representation. Unlike these approaches, CO-SSL only processes two images at each training iteration.

**Intra-network SSL.** A seminal work, AMDIM (Bachman et al., 2019), already proposed applying a contrastive loss between intermediate layers of a visual backbone with a limited RF. This approach improves visual representation learning (Hjelm et al., 2018; Bachman et al., 2019) and vision-based reinforcement learning agents (Anand et al., 2019; Mazoure et al., 2020). They use neither standard CNNs nor CNNs that have high-level representations with small RFs. We show in Section 4 that CO-SSL outperforms AMDIM. SDSSL (Jang et al., 2023) is similar to AMDIM, but uses all intermediate layers and focuses on vision transformers. This means that early layers quickly have a wide RF.

## 3 METHOD

CO-SSL aims to learn similar representations for visual features that co-occur in the same image. Instead of leveraging co-occurrences between image crops, we apply the loss function of a given SSL algorithm between local representations (before the final pooling layer) and the global representation of an image (last layer of the visual backbone). To analyze the impact of the RF size of local representations, we introduce a new family of convolutional architectures named Receptive Field ResNet (RF-ResNet). Each RF-ResNet is configured by the maximum RF size of its local representations, while keeping the same amount of layers and parameters.

### 3.1 CO-SSL: CO-OCCURRENCE SELF-SUPERVISED LEARNING

**CO-SSL in general.** In principle, CO-SSL is adaptable to most SSL methods (SimCLR Chen et al. (2020a), MoCoV3 Chen et al. (2021), BYOL Grill et al. (2020) . . . ). Relative to the original SSL method, the CO-SSL variant extracts local representations (before the average pooling layer) and individually forwards them to a new projection head. Then, CO-SSL computes the averaged SSL loss function between individual local embeddings and the global embedding computed by the original method. To demonstrate the generality of CO-SSL, we implement CO-BYOL, CO-MoCoV3 and CO-DINO. Because of space constraints we only describe CO-BYOL in detail.

**CO-BYOL in detail.** CO-BYOL simultaneously maximizes the standard BYOL loss between two global augmented image representations and a BYOL loss between local representations of an augmented image and the global representation of a differently augmented image. Figure 1 illustrates the learning architecture. As in the original BYOL, CO-BYOL includes an online network with learnable weights $\theta$, composed of the visual backbone $f_\theta$, a projection head $g_\theta^1$ and prediction head $q_\theta^1$. Another target network mirrors the online network, with weights $\xi$ defined as an exponentially moving average of the weights of the online network (Grill et al., 2020). To process local representations, we introduce two new projection heads $g_\theta^2$, $g_\xi^2$ and one prediction head $q_\theta^2$, all based on $1 \times 1$ convolution layers.

Let $x$ be an image uniformly sampled from a dataset $\mathcal{D}$, we apply two augmentations sampled from two distributions $t \sim \mathcal{T}$ and $t' \sim \mathcal{T}'$, which results in two views $v = t(x)$ and $v' = t'(x)$. The first view $v$ feeds an online network $f_\theta$, which outputs a set of local representations $L_\theta$ before the average pooling layer and a global image representation $h_\theta = f_\theta(v)$. The target network processes $v'$, resulting in another set of local representations $L'_\xi$ and a global representation $h'_\xi$. We apply the standard BYOL loss by computing the online prediction $p_\theta = q_{\mathsf{g},\theta}^1(g_\theta^1(h_\theta))$, the target projection $z'_{\mathsf{g},\xi} = g_\xi^1(h')$ and minimizing

$$\mathcal{L}_g(p_{\mathsf{g},\theta}, z'_{\mathsf{g},\xi}) = -2 \cdot \texttt{cosine}(p_{\mathsf{g},\theta}, sg(z'_{\mathsf{g},\xi})),$$

where $\texttt{cosine}$ denotes the cosine similarity and $sg$ represents the stop gradient operation.

Our contribution lies in the second application of the loss function. We extract the $n^2$ local representations from the view $v$, i.e. $l_\theta^i \in L_\theta$ where $i \in \{0, ..., n^2\}$. These are the representations of the visual backbone, right before average pooling. Then, we compute the local prediction of each of them $p_{1,\theta}^i = q_\theta^2(g_\theta^2(l_\theta^i))$. We similarly compute target local embeddings $z'^i_{1,\xi} = g_\xi^2(l'^i_\xi)$. We use $1 \times 1$ convolutions to parallelize the computations. Finally, we minimize the BYOL loss function between local and global embeddings:

$$\mathcal{L}_l(L_\theta, p_{\mathsf{g},\theta}, L'_\xi, z'_{\mathsf{g},\xi}) = -\frac{2}{n^2} \sum_{i \in \{0,...,n^2\}} \texttt{cosine}(p_{1,\theta}^i, sg(z'_{\mathsf{g},\xi})) + \texttt{cosine}(p_{\mathsf{g},\theta}, sg(z'^i_{1,\xi})),$$

where $p_{1,\theta}^i$ and $z'^i_{1,\xi}$ derive from $L_\theta$ and $L'_\xi$, respectively. In practice, we similarly process all views by both the online and target networks, and apply the losses accordingly (like in (Grill et al., 2020)). Overall, our loss function is:

$$\mathcal{L}_{\texttt{CO-BYOL}} = \mathcal{L}_g(p_{\mathsf{g},\theta}, z'_{\mathsf{g},\xi}) + \mathcal{L}_g(p'_{\mathsf{g},\theta}, z_{\mathsf{g},\xi}) + w_s \big[\mathcal{L}_l(L_\theta, p_{\mathsf{g},\theta}, L'_\xi, z'_{\mathsf{g},\xi}) + \mathcal{L}_l(L'_\theta, p'_{\mathsf{g},\theta}, L_\xi, z_{\mathsf{g},\xi})\big],$$

where $w_s$ is a hyperparameter ruling the trade-off between $\mathcal{L}_g$ and $\mathcal{L}_l$.

## 3.2 RF-RESNET FAMILY

To promote the learning of relationships among *local* image features, we want to avoid excessively large receptive field sizes prior to the final average pooling. Modern CNN architectures learn local representations with RF size beyond $400 \times 400$ pixels (Araujo et al., 2019). Given that the standard input images in SSL have a size of $224 \times 224$, each "local" representation may potentially be a representation of the whole image. Thus, we here propose a modification to the ResNet family to compute local representations with RF smaller than the size of the image. For a fair comparison, we further ensure that, for a given RF size, the number of layers and parameters remain the same. Note that optimizing the neural network architecture is orthogonal to our contribution (Tan & Le, 2019; Sandler et al., 2018). The RF-ResNet family, shown in Figure 2, includes three modifications compared to ResNets.

First, we reduce the RF size of ResNet's local representations (approximately $425 \times 425$ for a ResNet50). To achieve this, we severely limit convolution strides and kernel sizes (cf. Appendix B.1 for a formula to compute RF sizes, taken from Araujo et al. (2019)). We replace all $3 \times 3$ convolution layers by $1 \times 1$ convolution layers (with stride 1), except the first $3 \times 3$ convolution in the first and middle blocks in each of the four stacks of blocks. We also keep identical the very first convolution of the ResNet. The stride of the first $3 \times 3$ convolution in each middle block is always set to 1. Depending on the desired RF size, we add hyperparameters that control whether we keep the MaxPool layer ($m$) and denote the strides of the $3 \times 3$ convolution in the second to fourth blocks ($s$,

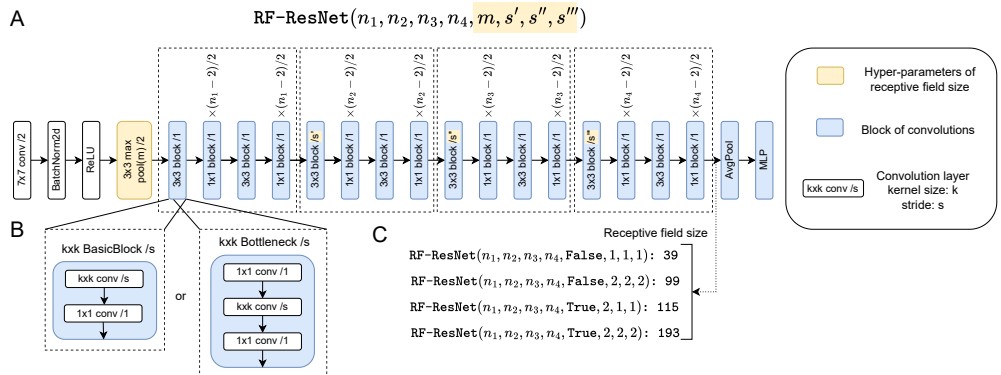

Figure 2: RF-ResNet architecture. We omit residual connections for better readability. A) Overview of the architecture as a succession of convolutions blocks. "n" is the number of blocks in each of the four layers as in a standard ResNet, $m$ denotes the absence/presence of the MaxPool layer and $s'$,$s''$,$s'''$ denote the values of the stride parameters of the first block of a layer (cf. B). B) Zoom in on the two different types of blocks, which are a stack of convolution layers. C) Examples of RF sizes for four different RF-ResNet; it is independent from the number of blocks $n > 2$ in each layer.

$s'$, $s''$). We also keep usual hyperparameters ($n_1$, $n_2$, $n_3$, $n_4$) to define the number of blocks in each layer.

Second, local features on their own may be insufficient to build strong global representations. Between the average pooling layer and the projection heads, we include an MLP with the same structure as one convolution block (BottleNeck or Basic depending on the ResNet), with a number of output units equal to the number of input units. To prevent this MLP from discarding information, as often happens with fully connected projection heads (Chen et al., 2020a), we keep skipped connections. In practice, we also evaluate the representation before the MLP, which corresponds to an averaged bag of patch representations.

Third, we cannot fairly compare an RF-ResNet to a ResNet because replacing $1 \times 1$ convolutions with $3 \times 3$ convolutions significantly reduces the number of parameters (Pan et al., 2022). Thus, for RF-ResNet18, we double the number of convolution blocks to get closer to the number of parameters of ResNet18 (8M vs. 11M); for RF-ResNet50, we add one block in the fourth layer (23.7M vs. 23.5M). Note that adding $1 \times 1$ convolution layers does not change the RF size of representations in the network. In the following, we call "RF99-ResNet50" a RF-ResNet50 with hyperparameters that encode local representations with a maximum RF of size $99 \times 99$.

## 3.3 TRAINING AND EVALUATION

We implemented CO-BYOL and CO-DINO in the solo-learn framework (Da Costa et al., 2022). We run CO-SSL on four datasets to assess its robustness and sample effiency, namely ImageNet-1K, ImageNet-1K with 10% of training samples, Tiny-ImageNet and ImageNet-100. For ImageNet-100, we follow finetuned hyperparameters provided in the public solo-learn codebase. For ImageNet-1K, Tiny-ImageNet, ImageNet-100, we apply hyperparameters provided in the original papers for ImageNet-1K. As an exception, we noticed that a projection layer with two hidden layers works better for BYOL-based methods on ImageNet-1K (100 epochs). We further finetune the initial learning rate for all methods in $\{0.2, 0.4, 0.8, 1.6\}$. For CO-BYOL and CO-DINO, we apply the same augmentations as in BYOL (Grill et al., 2020), except that we set the default minimum crop ratio to $c_{min} = 0.2$ (cf. section 4 for an analysis). In ImageNet-100 and ImageNet-1K, our default settings use a ResNet50 for baselines and RF99-ResNet50 for CO-SSL. We hyperparameterized the RF sizes on ImageNet-100 (cf. Section 4.3) and found RF99-ResNet50 to be the best for large image sizes ($224 \times 224$). In Tiny-ImageNet, we use ResNet18 for baselines and RF29-ResNet18 for CO-SSL, with the modifications on the very first layers to work on small images (He et al., 2016). We chose RF29-ResNet18 because it corresponds to a similar image portion in $64 \times 64$ as RF99-ResNet50 for $224 \times 224$ images: $(\frac{29}{64})^2 \approx (\frac{99}{224})^2 \approx 0.2)$. For CO-SSL, we select the best value

| Method | Top-1 ImageNet-1k accuracy |
|---|---|
| SimCLR (pub) (Chen et al., 2020a) | 66.5 |
| AMDIM (150e) (Bachman et al., 2019) | 68.1 |
| SimSiam (pub) (Chen & He, 2021) | 68.5 |
| MoCo-v2 (pub) (Chen et al., 2020b) | 67.4 |
| VICReg (pub) (Bardes et al., 2022a) | 68.6 |
| BYOL (pub) (Grill et al., 2020) | 69.3 |
| DINO+ (Caron et al., 2021) | 69.5 |
| BYOL+ (Grill et al., 2020) | 70.1 |
| MEC (Liu et al., 2022b) | 70.6 |
| Matrix-SSL (Zhang et al.) | 71.1 |
| **CO-BYOL (RF99-R50, patch)** | **71.2** |
| **CO-BYOL (R50)** | **71.4** |
| **CO-BYOL (RF99-R50)** | **71.5** |

Table 1: Top-1 linear validation accuracy (in %) of models pre-trained for 100 epochs on ImageNet-1K. For RF99-R50, "patch" means that we evaluate the representation right after average pooling, *i.e.* this is a simple averaged bag of patch representations. (pub) indicates published results reported in (Chen & He, 2021; Ozsoy et al., 2022; Liu et al., 2022b) and "+" denotes improved reproduction vs. published results.

of $w_s$ in $\{0.2, 0.5\}$ and analyze the role of this hyperparameter in Section 4.5. We will release the precise configuration files with the code upon acceptance.

To evaluate the methods, we train a linear probe online on top of the representations and report its final validation accuracy, following previous works (Bordes et al., 2023; Da Costa et al., 2022). For RF-ResNet, we evaluate the representations both before and after the MLP to compare the performance of an averaged bag of patch representations (before) versus a non-linear transformation of this bag (after).

## 4 EXPERIMENTS

Here, we first evaluate CO-SSL's representations on downstream categorization and their robustness to corruptions. Then, we analyze the factors contributing to its performance.

### 4.1 CO-BYOL OUTPERFORMS PREVIOUS METHODS ON DOWNSTREAM CATEGORY RECOGNITION

Here, we aim to evaluate the benefits of learning local representations that model spatial statistical co-occurrences. Table 1 shows the top-1 accuracy on ImageNet-1K of CO-BYOL after 100-epochs pre-training. CO-BYOL clearly outperforms all comparison baselines based on CNNs and surpasses BYOL by $1.4\%$. Interestingly, even with large receptive fields (R50, $425 \times 425$) or a simple averaged bag of patch representations (RF99-R50, patch), CO-BYOL outperforms previous methods.

In Table 2, we further evaluate the category recognition abilities of CO-SSL on ImageNet-100, Tiny-ImageNet and ImageNet-1K with $10\%$ and $100\%$ of the training data. We compare CO-SSL with their simple SSL counterparts and their multicrop version (SSL-mc) (Caron et al., 2021). We compare to SSL-mc because they arguably learn to extract visual co-occurrences; however, please note that the multicrop version unfairly uses several times more images per pre-training iteration than CO-SSL (cf. Appendix A.4 for a deeper analysis). Due to memory limitations, SSL-mc uses 6 small crops in Tiny-ImageNet and 4 small crops in other datasets (in addition to the 2 large crops). We tried the two usual ranges of multi-crop scales ($[0.05 - 0.14; 0.14 - 1]$ and $[0.14 - 0.4; 0.4 - 1]$) and took the best one. We observe that CO-BYOL outperforms all other methods, except DINO-mc in Tiny-ImageNet. CO-DINO also surpasses DINO, emphasizing the generality of CO-SSL. However, CO-DINO does not reach the performance of either of DINO-mc, or CO-BYOL. The reason is unclear to us, but we suspect that the design choices made in the original paper may favor DINO-mc (bottleneck layer, deep projection heads, normalizing the last layer...) and leave to future

Table 2: Top-1 linear validation accuracy with models trained on three different datasets. We use a RF99-ResNet50 with CO-BYOL. "mc" denotes the use of multicrop and "Epochs" refers to pre-training epochs. X% indicates the proportion of images used during pretraining and finetuning. We ran all numbers, they match or outperform published results.

|  | Epochs | BYOL | BYOL-mc | **CO-BYOL** | DINO | DINO-mc | **CO-DINO** |
|---|---|---|---|---|---|---|---|
| I-100 | 400 | 83.5 | 84.7 | **87.6** | 84.1 | 83.5 | 86.4 |
| Tiny-I | 400 | 51.3 | 56.1 | 56.6 | 55.1 | **57.4** | 57.3 |
| I-1K 100% | 100 | 70.1 | 70.9 | **71.5** | 69.5 | 70.9 | 70.3 |
| I-1K 10% | 300 | 46.1 | 52.5 | **53.4** | 49 | 50.9 | 49.8 |

Table 3: Corruption accuracies of models pre-trained and linearly finetuned on ImageNet-1K (100 epochs) against ImageNet-C corruptions. Corruption accuracies are averages of accuracies across five degrees of corruption severity.

|  | Noise | | | Blur | | | | Intern. |
|---|---|---|---|---|---|---|---|---|
|  | Gaus. | Shot | Imp. | Defoc. | Glass | Motion | Zoom | Mask |
| BYOL | 28.5 | 26.7 | 19.2 | 35.1 | 21.1 | **29.6** | 24.6 | 54.5 |
| BYOL-mc | 26.4 | 24.2 | 12.6 | 31.3 | 16.9 | 27.2 | 24.3 | 53.9 |
| CO-BYOL (R50) | **34.5** | **33.5** | **24.7** | **35.1** | **21.2** | 28.5 | **24.9** | 60.4 |
| CO-BYOL (RF99-R50) | 19.6 | 18.9 | 8.0 | 27.5 | 16.4 | 24.2 | 24.2 | **65.5** |

work an analysis of how these designs impact CO-DINO. Overall, we conclude that in almost all of the studied settings CO-BYOL learns better visual representations than previous methods.

## 4.2 CO-SSL (RESNET50) IS MORE ROBUST TO CORRUPTIONS AND ADVERSARIAL ATTACKS

In this section, we investigate whether CO-SSL leads to more robust global representations. We focus on BYOL variants and provide results for DINO variants in Appendix A.7. First, we assess the robustness of our pre-trained models to ImageNet-C corruptions (Hendrycks & Dietterich, 2018). We focus on blur and noise corruptions, please, see (Hendrycks & Dietterich, 2018) for details. In Table 3, the results consistently indicate that global representations of CO-BYOL (R50) are consistently more robust to noise corruptions. However, we do not see clear differences with respect to blur-based corruptions. We further investigate the robustness of global representation with respect to internal corruptions. To this end, we randomly discard local representations before computing global representations and compute the category recognition accuracy. We test the removal of $90\%, 80\%, 70\%, 60\%, 50\%$ of local representations and average the results. Table 3 (Mask) clearly shows that CO-BYOL is more robust to internal corruptions. CO-BYOL (RF99-R50) performs the best in this task, but we can not rule out the possibility that it comes from its higher number of local representations ($14 \times 14$).

Then, we study whether CO-BYOL is more robust to small adversarial attacks, although the model is not designed to be adversarially robust. We focus on Projected Gradient Descent (PGD) attacks ($\mathcal{L}_{inf}$) (Kurakin et al., 2018), implemented with Foolbox (Rauber et al., 2017). PGD iteratively 1- backpropagates the negative gradient of the classification loss on the image and 2- modifies the image with the clipped image gradient. We refer to (Kurakin et al., 2018) for more information on PGD. In Table 3, we find that CO-BYOL (R50) consistently outperforms its BYOL counterpart. CO-BYOL (RF99-R50) does not present advantages with respect to adversarial attacks or image corruption. We suspect this may be due to the difference of neural architecture. Overall, CO-BYOL with a ResNet-50 leads to more robust visual representations.

## 4.3 CO-SSL IMPROVES OVER C/R AT MODELING SPATIAL CO-OCCURRENCES

Here, we assess the potential of modeling statistical co-occurrences using local representations (through RF size) rather than small images (through the minimum crop ratio). In Figure 3A), we observe that CNNs with RF sizes between $67 \times 67$ and $163 \times 163$ yield better results with CO-BYOL

Table 4: Adversarial robustness of models pre-trained and linearly finetuned on ImageNet-1K (100 epochs) against PGD attacks. For an attack: $\epsilon$ defines the perturbation distance, $\gamma$ the step size and "Iterations" the number of steps.

| $\epsilon$ | 0.003 | 0.01 | 0.03 | 0.1 | 0.003 | 0.01 | 0.003 | 0.01 |
|---|---|---|---|---|---|---|---|---|
| $\gamma$ | $\epsilon/40$ | $\epsilon/40$ | $\epsilon/40$ | $\epsilon/40$ | $\epsilon/40$ | $\epsilon/40$ | $\epsilon/10$ | $\epsilon/10$ |
| Iterations | 1 | 1 | 1 | 1 | 5 | 5 | 1 | 1 |
| BYOL | 64.7 | 54.1 | 34.7 | 8 | 45.1 | 14.4 | 51.7 | 27.4 |
| BYOL-mc | 64.4 | 50.8 | 26.4 | 3.3 | 39.9 | 7.7 | 47.5 | 18.6 |
| CO-BYOL (R50) | **67.4** | **59.4** | **43.9** | **15.4** | **52.0** | **23.1** | **57.2** | **36.3** |
| CO-BYOL (RF99-R50) | 63.8 | 49.2 | 27.0 | 4.3 | 37.1 | 7.5 | 18.6 | 3.8 |

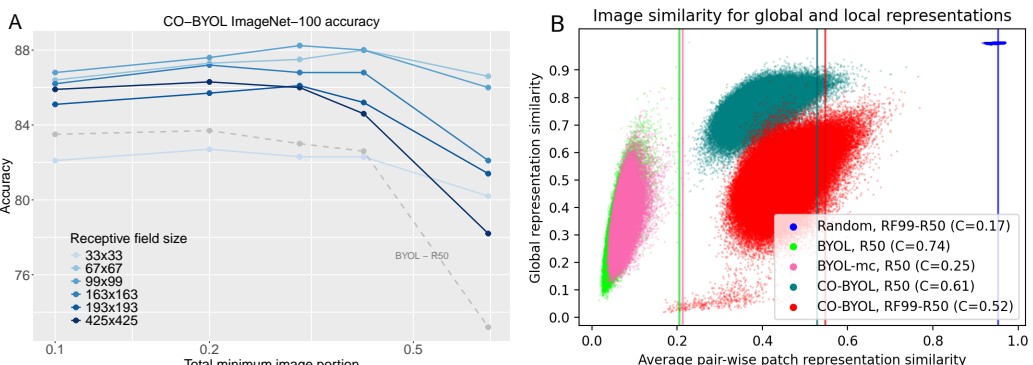

Figure 3: A) Top-1 ImageNet-100 validation accuracy. We train different RF-ResNet50 and different minimum crop ratios $c_{min} \in \{0.1, 0.2, 0.3, 0.4, 0.7\}$. Due to its design, it is impossible to reach a RF of size $33 \times 33$ with a RF-ResNet, we use a parameter-matched BagNet33 (Brendel & Bethge, 2018). BagNets are similar to RF-ResNet, but defined for smaller ranges of RFs. For size(RF) = $425 \times 425$, we use a ResNet50. We also show BYOL with ResNet-50, as a reference baseline. B) Correlation between the cosine similarity between global representations and the cosine similarity between local representations on ImageNet-1K validation set. "C" denotes the Pearson correlation and vertical bars shows the cosine similarity between intra-image local representations.

than tiny ($33 \times 33$) or large ($425 \times 425$ and $193 \times 193$) RF sizes. This performance gap is particularly enhanced for large minimum crop ratios ($c_{min} = 0.7$), suggesting that CO-BYOL with small RFs can replace C/R, to some extent. A relatively low $c_{min}$ remains important, presumably because C/R also has a regularization effect Hernández-García & König (2018). Overall, we conclude that CO-SSL with small RF sizes improves over C/R at modeling spatial co-occurrences.

To further investigate why CO-SSL is better than C/R at modeling spatial co-occurrences, we study the impact of two hyperparameters ruling the number and importance of spatial co-occurrences used by CO-SSL in Table 5. First, we observe that a good value for the weight coefficient ranges in $[0.2, 0.5]$. Second, we also assess the impact of applying the local loss function $\mathcal{L}_l$ only on a subset of local representations at each iteration. This speeds up the training process with wide/deep projection heads, but decreases the number of co-occurring representations that are made similar at each iteration. To do so, we spatially downsample the feature maps before feeding the local projection heads $g^2$. We see that the accuracy decreases slowly when exponentially decreasing the size of the feature maps. We conclude that CO-BYOL benefits from making similar many co-occurring image patches per sampled image. This partly explains its better performance with respect to C/R, which uses only one pair of co-occurring image subparts.

Table 5: Top-1 linear validation accuracy (in %) when trained on ImageNet-100 with different hyperparameters. CO-BYOL uses a RF99-ResNet5°. We consider the weight coefficient $w_s$ and the number of spatial co-occurrences per sampled image $n^2$ to which CO-SSL applies $\mathcal{L}_l$. By default, we use the maximum ($n^2 = 196$) for CO-BYOL with RF99-ResNet50. We also show BYOL with ResNet50.

| | BYOL | CO-BYOL | | | | | BYOL | CO-BYOL | | | | |
|---|---|---|---|---|---|---|---|---|---|---|---|---|
| $w_s$ | 0 | 0.1 | 0.2 | 0.5 | 1 | $n^2$ | 0 | 1 | 4 | 16 | 64 | 196 |
| Acc | 83.9 | 84.7 | 87.6 | 88 | 87 | Acc | 83.9 | 84.2 | 86.5 | 87.3 | 87.7 | 87.6 |

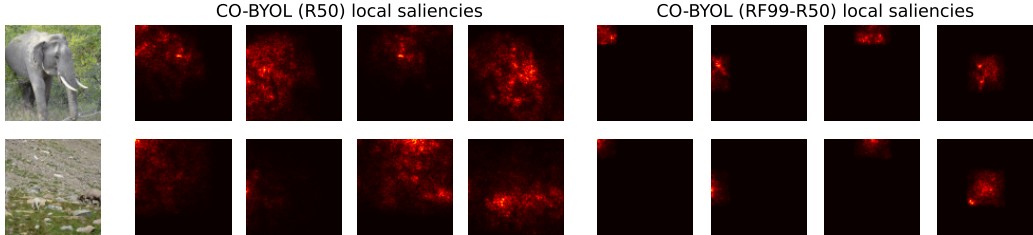

Figure 4: Visualization of effective receptive fields of 4 local representations computed on two validation images for two methods. For diversity, we select four local representation with normalized coordinates in the features maps (0,0), (0, 0.5), (0.5,0) and (0.5,0.5) from left to right.

### 4.4 CO-SSL BUILDS SIMILAR LOCAL REPRESENTATIONS BY INFERRING STATISTICAL CO-OCCURRENCES.

To better understand the impact of CO-BYOL on the learnt representations, we extract a random subset of 250,000 different image pairs from the ImageNet-1K validation set and compute the cosine similarity between global representations of images and the average pair-wise cosine similarity between local representations of images. First, we observe in Figure 3B) that local similarities strongly correlate with global similarities for all methods and architectures, confirming that local representations shape the structure of distances between global representations. Second, an important difference between CO-BYOL and BYOL is the increased similarity between intra-image local representations (vertical bars). This indicates that CO-BYOL extracts similar local representations within an image, regardless of the neural architecture. This is likely due to the triangle inequality: local representations of an image are all pushed towards the same global representation, and thus pushed towards each other. We presume that this explains the robustness of CO-SSL, as a corruption must then alter several local representations instead of one to corrupt a global feature.

The high similarity between local representations may have two origins. First, it may be that CO-BYOL infers co-occurrence statistics to construct a features representation that is invariant to frequently co-occurring features representations. Second, CO-BYOL may shortcut the learning process by learning the same local representations covering the whole image, but through different paths in the visual backbone. The latter point is impossible for RF99-R50 due to the bounded receptive fields of local representations; this means that CO-BYOL indeed infers their co-occurrence statistics. To investigate the strategy of CO-BYOL (R50), we plot the saliency maps resulting from maximizing all the units in local representations in Figure 4. This allows us to see the *effective* RFs of local representations for a given image (Luo et al., 2016). We observe that the saliency maps of CO-BYOL (R50) cover distinct subparts of an image, although larger than CO-BYOL (RF99-R50). Furthermore, our quantitative analysis in Appendix A.5 indicates that the number of "impactful" pixels is only slightly higher for a ResNet50 than for RF99-ResNet50. Thus, CO-BYOL (R50) probably employs a mixed strategy of shortcut and inference. This offers an explanation of why CO-BYOL (RF99-50) performs slightly better than CO-BYOL (R50) in Table 1, 6 and 7. Overall, CO-BYOL manages to construct (dis)similar local representations by inferring their frequency of co-occurrences.

Table 6: Top-1 linear validation accuracy (in %) of models trained on ImageNet-100. In RF-ResNetv0, we remove the MLP after the average pooling of RF-ResNet. (patch) indicates evaluation right after average pooling. (1 head) denotes the use of the same head to compute local and global embeddings of CO-BYOL, we take the best layer between before and after the MLP.

|         | R50  | RF99-R50v0 | RF99-R50 | RF99-R50 (patch) | RF99-50 (1 head) |
|---------|------|------------|----------|------------------|------------------|
| BYOL    | 83.5 | 80.7       | 83.9     | 83               | 83.9             |
| CO-BYOL | 86.3 | 87.7       | 87.5     | 87.6             | 87.5             |

### 4.5 ABLATION STUDY

Table 6 shows an ablation study of CO-BYOL. We observe that CO-BYOL boosts both the original ResNet50 and the RF-ResNet50 (BYOL versus CO-BYOL), but this boost is slightly stronger for RF-ResNet. Interestingly, the MLP of RF-ResNet shows an important effect only when CO-BYOL is absent (RF-ResNet versus RF-ResNetv0). One may hypothesize that this comes from the MLP simulating an extra layer in the projection head. However, the accuracy after/before this layer remains similar (RF-ResNet50 versus RF-ResNet50 pool). The difference is similar on ImageNet-1K with 100 epochs (71.5% vs. 71.2% in Table 1). We also find that CO-BYOL (RF99-ResNet50v0) outperforms Co-BYOL (ResNet50), despite using fewer parameters (21.4M vs. 23.5M). In sum, CO-BYOL is essential for learning a high-quality bag of patch representations.

## 5 CONCLUSION

We proposed CO-SSL, a family of SSL approaches that make similar local representations (before final pooling layer) and global representations (before projection heads). We instantiated two models of this family, namely CO-BYOL and CO-DINO. We applied CO-SSL with standard CNNs and our newly introduced RF-ResNet, a CNN family that bounds the size of the receptive field of local representations. We tested CO-SSL on a series of datasets, including ImageNet-1K (100 epochs), and found that CO-BYOL consistently outperforms the comparison baselines. We also discovered that CO-BYOL is more robust than BYOL to noise corruptions, internal masking, small adversarial attacks and large minimum crop ratios. Our analysis demonstrates that CO-SSL assigns similar local representations to co-occurring visual features, which presumably rules the advantages of CO-SSL.

We found that a simple averaged bag of patch representations (20% of the image) learnt by CO-BYOL reaches 71.2% of accuracy on ImageNet-1K (100 epochs), which is on par with CO-BYOL (ResNet-50). Such a good performance seems counterintuitive, as it may critically prevent the integration of local features into global ones. However, recent findings highlight that standard CNNs mostly extract similar patchworks of local features without taking into account their global arrangement (Baker & Elder, 2022; Baker et al., 2018; Jarvers & Neumann, 2023). In humans, the ventral stream may also employ this strategy (Jagadeesh & Gardner, 2022; Ayzenberg & Behrmann, 2022), emphasizing the potential importance of this processing stage. Yet, we lacked computational resources to touch the boundaries of what can be learned with a bag of small patch representations. That would require even deeper ResNets and more training epochs.

Previous works highlighted the importance of different real-world co-occurrences to build semantic visual representation, e.g. across modalities (Radford et al., 2021) or time (Aubret et al., 2023; Parthasarathy et al., 2023). Here, we also showed the advantages of modelling spatial co-occurrences at the level of the whole object and large object parts through local and global representations. In theory, one could exploit even lower-level visual co-occurrences (e.g. between neighboring pixels or low-level representations) for learning intermediate representations. Given our results, we speculate that this may make the representations even more robust to corruptions and adversarial attacks.

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

Table 7: Top-1 linear validation accuracy (in %) of models pre-trained for 300 epochs on ImageNet-1K. (pub) indicates published results reported in (Chen & He, 2021; Ozsoy et al., 2022; Liu et al., 2022b).

| Method | Pre-training epochs | Accuracy |
|---|---|---|
| SimCLR (Chen et al., 2020a) | 300 | $\approx 67.5$ |
| SimSiam (pub) (Chen & He, 2021) | 400 | 70.8 |
| MoCo-v2 (pub) (Chen et al., 2020b) | 400 | 71 |
| BYOL (ours) (Grill et al., 2020) | 300 | 72.4 |
| BYOL (Grill et al., 2020) | 300 | 72.5 |
| **CO-BYOL (R50)** | 300 | 72.9 |
| MEC (Liu et al., 2022b) | 400 | 73.5 |
| Matrix-SSL (Zhang et al.) | 400 | 73.6 |
| **CO-BYOL (RF99-R50)** | 300 | **73.9** |

Table 8: Top-1 linear validation accuracy with models trained on three different datasets. "mc" denotes the use of multicrop and "Epochs" refers to pretraining epochs. X% indicates the proportion of images used during pretraining and finetuning.

| | Epochs | MoCoV3 | **CO-MoCoV3** |
|---|---|---|---|
| I-100 | 400 | 81.8 | **84.4** |
| Tiny-I | 400 | 51.78 | **56.02** |
| I-1K 100% | 100 | 69.8 | **70.2** |

## A  ADDITIONAL EXPERIMENTS

### A.1  ADDITIONAL IMAGENET RESULTS

**ImageNet-1k, 300 epochs.**  In Table 7, we compare CO-BYOL with other methods when pre-training for 300 epochs on ImageNet-1k. Here, we train a linear classifier on top of frozen representations for 100 epochs. We use an initial learning rate of 0.1, which is divided by 10 at epochs 60 and 80. As augmentations, we use horizontal flip and Crop/Resize with a minimum crop of size 8%.

Our results confirm the conclusions made in section 4.1, as CO-BYOL outperforms other methods for category recognition using an equal or smaller number of epochs. Interestingly, there is a bigger gap between CO-BYOL (R50) and CO-BYOL (RF99-R50) than when pre-training with 100 epochs. This further suggests that CO-BYOL better leverages spatial co-occurrences with longer training epochs when using a RF99-ResNet50. We presume this is due to the smaller size of their ERFs (cf. section 4.4).

**CO-MoCoV3.**  In Table 8, we show results for a CO-SSL variant of MoCoV3 Chen et al. (2021), namely CO-MoCoV3. We used the same hyperparameters as BYOL and CO-BYOL and a temperature of 0.2. We observe that CO-MoCoV3 consistently outperforms MoCoV3, in line with the results discussed in Section 4.1.

### A.2  ADDITIONAL TRANSFER LEARNING RESULTS

In this section, we train a linear probe on top of our I-1K models to evaluate additional properties of CO-BYOL. First, we evaluate whether better features on I-1K are also better for a scene recognition dataset like Places365-standard Zhou et al. (2017). In Table 9, we observe that CO-BYOL (RF99-50) outperforms BYOL by almost 1%. CO-BYOL (R50) does not work as well, which may be due to the effect described in section 4.4. Overall, this suggests that modeling spatial co-occurrences with CO-SSL can also boost scene recognition.

Next, we evaluate intermediate layers of ImageNet-1k pre-trained models (100 epochs) with an offline linear probe, following Wang et al. (2025). Table 10 shows that the last layer of the visual

Table 9: Top-1 validation accuracy on Places365-standard after training a linear probe on top of I-1K pre-trained models. We use the same linear training as in Appendix A.1.

|  | BYOL | CO-BYOL (R50) | CO-BYOL (RF99-R50) |
|---|---|---|---|
| Places365 acc | 49.8 | 49.6 | **50.7** |

Table 10: Top-1 validation accuracy on ImageNet-1k after 100 pre-training epochs. Here, we use the same setting for training a linear classifier as in Appendix A.1. "Final" refers to the last layer of the visual backbone, *i.e.* the same layer used in Table 1, but evaluated with an offline linear probe.

| Layer | BYOL | CO-BYOL (R50) | CO-BYOL (RF99-R50) |
|---|---|---|---|
| Layer 3 | 52.6 | **60.4** | 51.4 |
| Layer 4 | 64.1 | 66.1 | **68** |
| Final | 69.7 | 71.9 | **72** |

backbone is the best for category recognition, for all models. This observation is consistent with previous observations Wang et al. (2025).

### A.3 Optimal total minimum image portion

Here, we further assess the potential of modeling statistical co-occurrences using local representations (through small RF size $RF_s$) rather than small images (through small minimum crop ratio $c_{min}$). To this end, we compute the total minimum image portion used to create visual co-occurrences as $T_{min} = \max(1, \frac{RF_s^2}{I_x \times I_y}) \times c_{min}$, where $I_x$ and $I_y$ are the width and height of input images, respectively. In Figure 5, we plot again Figure 3A) with the total minimum image portion on the x-axis. We clearly see that the optimal total minimum image portion lies around 5% of the image. However, reducing the minimum crop ratio alone beyond 10% (far left point in each line) always decreases the accuracy, suggesting that small RFs are crucial to benefit from co-occurrences with very small image portions. Overall, This confirms the conclusion made in Section 4.3, *i.e.* CO-SSL is more efficient at modeling spatial co-occurrences than the crop/resize augmentation with two crops.

### A.4 CO-SSL is three times more sample-efficient than multi-crops

Here, we further assess the efficiency of CO-BYOL compared to BYOL-mc. We consider a batch size of 64 on one GPU, a ResNet50 (or RF99-ResNet50), 2 large augmented images per input images ($224 \times 224$) and 4 additional small ($96 \times 96$) augmented images for BYOL-mc. First, we count the

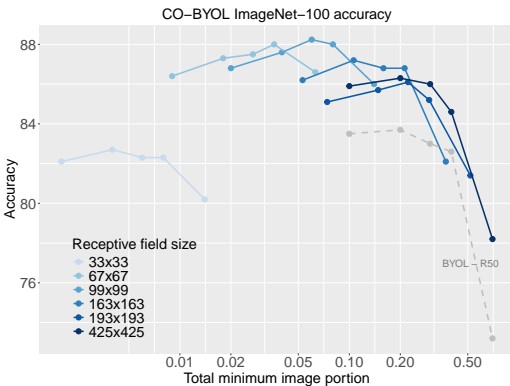

Figure 5: Top-1 ImageNet-100 validation accuracy. This is the same data shown in Figure 3, but we plot the accuracy against the minimum crop size.

Table 11: Comparison of the efficiency of the visual backbone between methods and CNNs architectures. We show ImageNet-1k accuracy for after pre-training for 100 epochs. There is no local heads in BYOL and BYOL-mc.

|  | BYOL | BYOL-mc | CO-BYOL (R50) | CO-BYOL (RF99-R50) |
|---|---|---|---|---|
| Backbone FLOPs (G) $\downarrow$ | **32.8** | 38.4 | **32.8** | 106 |
| Local head FLOPs (G) $\downarrow$ | **0** | **0** | 5.6 | 22 |
| Memory (GB) $\downarrow$ | **12** | 36 | **12** | 12.1 |
| Images/iterations $\downarrow$ | **2** | 6 | **2** | **2** |
| I-1k 100% acc $\uparrow$ | 70.1 | 70.9 | **71.4** | **71.5** |

number of floating point operations (FLOPs) in the visual backbone per training iteration. We also show the number of FLOPs for local heads (projection heads $g_\theta^2$ and $g_\xi^2$ and the prediction head $q_\theta^2$), each composed of one hidden layer 4096 neurons and an output layer of 256 neurons. The global heads used by all methods have marginal FLOPs ([0.1, 0.3]G). For BYOL-mc, we also add 4 small crops ($96 \times 96$). Second, we calculate the theoretical memory after the forward pass but before the backward pass. We compute this as the number of parameters multiplied by the number of input images and four (a "float32" is encoded with eight memory bytes).

In Table 11, we observe that CO-BYOL (RF99-R50) uses more FLOPs than other methods, because it internally computes larger feature maps. BYOL-mc uses three times more training images than other methods, which also comes with an extra requirement for memory. In contrast, CO-BYOL (R50) achieves a good trade-off of accuracy, memory, FLOPs and sample efficiency.

A.5 QUANTITATIVE ANALYSIS OF THE RF

In Section 4.4, we qualitatively observed that local representations of CO-BYOL (R50) focus on distinct subparts of an image. To investigate this question quantitatively, we compute the ERF of BYOL and CO-BYOL following a procedure similar to Luo et al. (2016) on the ImageNet-1k validation set: we backpropagate the gradient of local representations onto the image, normalize the resulting saliency map with its maximum value and compute the square root of the number of pixels with a gradient value superior to $1 - 95\% = 5\%$ and to $1 - 68\% = 32\%$ of the best gradient value.

Figure 6 shows that a ResNet50 has a much larger number of pixels that slightly impact local representations ($> 5\%$) compared to RF99-ResNet50. We also see that CO-BYOL tends to slightly extend the ERF, compared to BYOL. In contrast, if we look at the number of "important" pixels ($> 32\%$) in Figure 6, we observe that the difference of ERF sizes between a ResNet50 and a RF99-ResNet50 is relatively small. This suggests that a local representation is mostly shaped by very local visual features.

A.6 "LAYER4" IS THE BEST LAYER TO SAMPLE LOCAL REPRESENTATIONS

Previous work proposed to maximize a SSL loss between the final representation and intermediate representations Jang et al. (2023); Bachman et al. (2019). Here, we replicate their analysis with our RF99-ResNet, loss function, a crop size of $0.2$ and a downsampling of $64$ (because feature maps can be very large in intermediate representations). Following the ResNet nomenclature, we find that selecting CO-BYOL's local representations in the "layer3" or "layer2" of RF99-ResNet leads to a respective decrease of accuracy of $0.4\%$ and $2.1\%$ in ImageNet-100. We deduce that the last layer is more appropriate for CO-SSL.

A.7 ROBUSTNESS ANALYSIS WITH CO-DINO

In this section, we replicate the experiments of section 4.2 with DINO and CO-DINO to assess their robustness. In Table 12, we observe that CO-DINO (R50) is more robust to noise corruptions than DINO. Surprisingly, we find that CO-DINO is sensitive to internal masking, compared to DINO and CO-BYOL (cf. Table 3). We do not have a good reason for that. In Table 13, we also observe that CO-DINO (R50) is more robust to adversarial attacks than DINO. Overall, the results are consistent with CO-BYOL, highlighting the generality of CO-SSL.

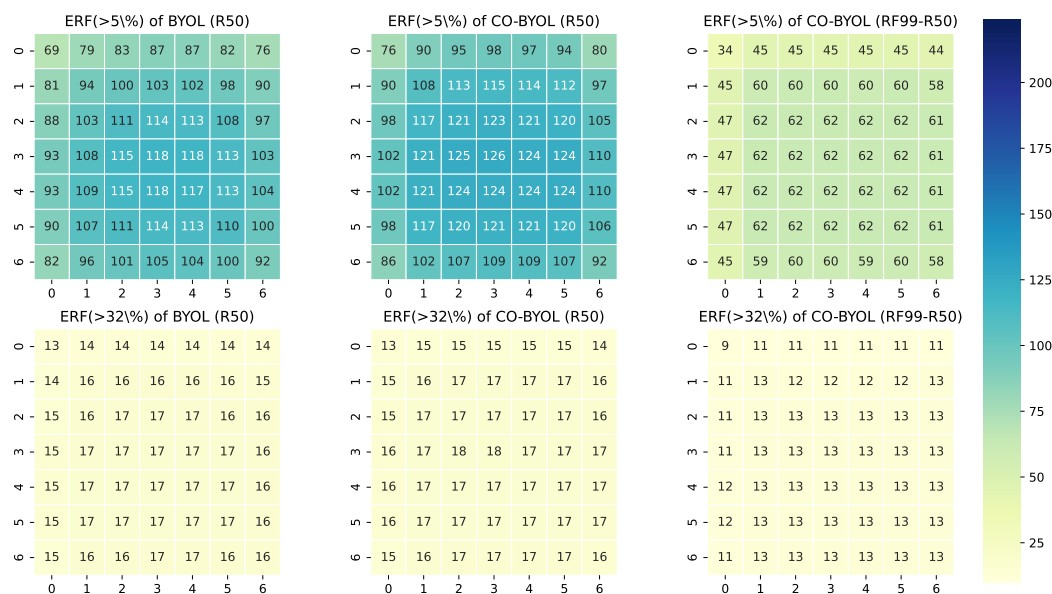

Figure 6: Effective receptive fields of local representations computed on ImageNet-1k validation set for different pre-trained models (100 epochs). We apply a stride of 2 to extract local representations of RF99-R50 to speed up the computations and observe the same number of local representations as R50.

Table 12: Corruption accuracies of models pre-trained and linearly finetuned on ImageNet-1K (100 epochs) against ImageNet-C corruptions. Corruption accuracies are averages of accuracies across five degrees of corruption severity.

|  | Noise | | | Blur | | | | Intern. |
|---|---|---|---|---|---|---|---|---|
|  | Gaus. | Shot | Imp. | Defoc. | Glass | Motion | Zoom | Mask |
| DINO | 7.6 | 8.2 | 4.7 | 14.1 | 11.7 | 10.0 | 9.5 | 30.4 |
| DINO-mc | 26.0 | 23.1 | 12.4 | **35.6** | 18.8 | **30.4** | **26.9** | **56.9** |
| CO-DINO (R50) | **31.0** | **29.7** | **22.4** | 33.9 | **20.6** | 28.5 | 24.6 | 37.6 |
| CO-DINO (RF99-R50) | 20.4 | 19.6 | 8.3 | 27.2 | 14.8 | 23.6 | 23.3 | 26.7 |

Table 13: Adversarial robustness of models pre-trained and linearly finetuned on ImageNet-1K (100 epochs) against PGD attacks. For an attack: $\epsilon$ defines the perturbation distance, $\gamma$ the step size and "Iterations" the number of steps.

| $\epsilon$ | 0.003 | 0.01 | 0.03 | 0.1 | 0.003 | 0.01 | 0.003 | 0.01 |
|---|---|---|---|---|---|---|---|---|
| $\gamma$ | $\epsilon/40$ | $\epsilon/40$ | $\epsilon/40$ | $\epsilon/40$ | $\epsilon/40$ | $\epsilon/40$ | $\epsilon/10$ | $\epsilon/10$ |
| Iterations | 1 | 1 | 1 | 1 | 5 | 5 | 1 | 1 |
| DINO | 38.3 | 32.5 | 20.6 | 3.7 | 28.7 | 11.4 | 31.2 | 17.2 |
| DINO-mc | **65.2** | 52.2 | 28.0 | 4.5 | 42.9 | 9.6 | 49.7 | 20.8 |
| CO-DINO (R50) | 64.9 | **55.9** | **38.4** | **10.7** | **48.1** | **18.5** | **53.5** | **31.2** |
| CO-DINO (RF99-R50) | 63.1 | 49.1 | 26.6 | 4.0 | 36.7 | 7.2 | 45.2 | 17.8 |

## B    ADDITIONAL DETAILS

### B.1    CALCULATION OF RF SIZE

For completeness, we provide the formula used to compute the size of receptive fields of a CNN at a given layer $L$ Araujo et al. (2019):

$$\text{RFS}(L) = \sum_{l=1}^{L} \left( (k_l - 1) \prod_{i=1}^{l-1} s_i \right) + 1,$$

where $k_l$ and $s_l$ are the kernel size and stride of a layer $l$, respectively. Overall, at each layer, a receptive field becomes larger by the product of all previous strides and the current kernel size. We further verified the receptive field sizes by computing a saliency map of the spatially central local representations of a random network and counting all non-zero values.

