# OpenReview forum: "Seeing the Whole in the Parts in Self-Supervised Representation Learning"
_ICLR.cc/2025/Conference — Submitted to ICLR 2025_

### Official Review · Reviewer_mcd1 · 2024-11-02

**Soundness:** 3
**Presentation:** 1
**Contribution:** 2
**Rating:** 3
**Confidence:** 4

**Summary:**

The paper studies the task of self-supervised learning and proposes a family of instance discrimination methods that make global and local representations similar, called COSSL. It shows improved performance over existing SSL methods across various settings of ImageNet.

**Strengths:**

- the paper is easy to follow
- the proposed method is simple yet effective, and it shows superior results across settings
- the authors show the proposed method can be integrated into other strong SSL methods (e.g. BYOL and DINO) and leads to further performance improvement. If this holds true across different architectures and SSL methods, it could potentially be an important technique in the SSL toolkit.
- the performance improvement on the robust benchmark is interesting and important.

**Weaknesses:**

- The presentation of the paper should be improved:
1. consider explicitly listing the contributions of the paper at the beginning.
2. improve the quality of the figures and the corresponding captions. The captions for Figures 1 and 2 do not help in understanding the method at all.
3. which part of Section 3.1 is specific to BYOL, and which parts are general and can be used in other methods?
4. avoid using the same notation to denote different things, for example, $z^{'i}_{\xi}$ is a different thing from the one without I.
5. avoid abusing terminologies. For example, the authors evaluate the robustness of the methods in Table 3. But Line 312 says Table 2 is also an evaluation of robustness. This would be understandable if there were no Table 3 and evaluation against adversarial examples. But in the current context, it leads to confusion.

- The method, at first glance, seems to be more general than what is in the paper. However, the current method has many specific designs. For example, the method is only evaluated on ResNet. I think it is quite natural to wonder how the proposed method behaves on more recent architectures such as transformers, MLP-mixers, etc. Should the loss look different on different architectures or it is optimal to minimize the negative cosine similarity? How should one define "global feature" and "local feature"  in other settings? For example, if one wants to deploy the method in transformers, should one use the CLS token as the global feature or the pooling of all patch features?

- What is the motivation behind the RF-ResNet? The paper says it is used for analysis purposes (line 211). But it is also mentioned as one of the contributions on line 64. What is the advantage of RF-ResNet compared to standard ResNet? From what I understood, RF-ResNets are less robust than the standard ResNet and do not bring significant performance improvement. Table 1 is the only place the authors show that RF-ResNet50 is slightly better than ResNet50. However, RF-ResNet 50 also has slightly more parameters than ResNet50 (23.7M vs 23.5M on line 238). So I'm not sure if it is a valid contribution besides the analysis purposes.

- How is the generalization of the learned representations from the proposed method to other downstream tasks beyond ImageNet classification? While linear probing on ImageNet is quite common for assessing the quality of representation learning methods, a good linear probing performance does not always translate to other tasks such as scene understanding. When an image describes a complicated scene, I'm not sure if it is really beneficial to force all local features to be close to the global feature because there could be more than one object/topic. Perhaps it makes more sense to have more than one global feature or set the number of global features as a hyper-parameter. For example, when using 4 global features, the final feature map can be divided into 4 regions and each region computes its own loss as in equation 2.

**Questions:**

- Does it make sense to also use the loss from Equation 2 at intermediate layers? I saw in A.3 that the authors did some initial experiments by selecting local features from intermediate layers. I wonder what if the global features are also from the same layer (otherwise, it does not make sense to use it as the "anchor point" for features from different layers).

- CO-DINO is also evaluated in many places in the paper. However, the author did not provide any implementation details in terms of how to combine CO-SSL with DINO.

- Is it necessary to use different projectors and predictors for computing the loss in equation 2? What does the performance look like if the local features are also from the same projectors and predictors as the global features?

- Why 99 and 29 are chosen as the RF for ResNet 50 and ResNet 18 in the experiments? Does the same conclusion hold for other RF sizes?

- What is the depth of the additional MLP added after the pooling layer? What is the effect of the depth of this MLP to the final performance?

---

> ### Author Response · Authors · 2024-11-26
> **Rebuttal [1/2]**
>
> We thank the reviewer for the careful reading. We answer each point below.
>
> > Consider explicitly listing the contributions of the paper at the beginning.
> We added a brief description of our contributions. We propose CO-SSL, a new family of SSL methods that model co-occurrences among local and global representations; we introduce a new CNN architecture, RF-ResNet, that extracts a bag of local patch repre-
> sentations to harness CO-SSL’s ability to align local and global representations; we identify which properties of CO-SSL make it better at category recognition and more robust to diverse corruptions and small adversarial attacks.
> > Improve the quality of the figure 1 and 2 and the corresponding captions.
>
> We extended the captions and completed the legends of the figures.
>
> > Which part of Section 3.1 is specific to BYOL, and which parts are general and can be used in other methods?
>
> We added a paragraph in Section 3.1 to clarify CO-SSL for other methods. To adapt a method named X, one needs to: 1) apply X without changes ; 2) add a new projection head that computes local embeddings from local representations before the average pooling layer; 3) compute the averaged loss function of X between each local embedding and the global embedding originally computed by X.
>
> > Avoid using the same notation to denote different things, e.g z^i versus z
>
> We updated z^i to z^i_l and z to z_g (and similarly for p). This clarifies that z_g denotes the embeddings of local representations while z_l denotes local embeddings. We decided to keep z (and p) as it is the canonical letter for denoting embeddings (and predictions) in SSL.
>
> > The authors evaluate the robustness of the methods in Table 3. But Line 312 says Table 2 is also an evaluation of robustness.
>
> Indeed, this was a misuse of the word “robustness”. We updated line 312 to “...evaluate the category recognition abilities…”.
>
> > How the proposed method behaves on more recent architectures such as transformers, MLP-mixers, etc. How should one define "global feature" and "local feature" in other settings? For example, if one wants to deploy the method in transformers, should one use the CLS token as the global feature or the pooling of all patch features?
>
> In this paper, we focus on convolutional architectures and show that CO-SSL can harness the small effective receptive fields of their representations before pooling. Different architectures have different inductive biases and we leave to future work experiments on Vision Transformers with the different ways to define a global representation. Please, also see our answer to Reviewer mnvG regarding vision transformers.
>
> > Is it optimal to minimize the negative cosine similarity?
>
> No, the use of cosine similarity for CO-BYOL is due to its use in BYOL. CO-DINO and CO-MoCoV3 use the loss function of DINO and MoCoV3, respectively. We clarified that in a new paragraph in Section 3.1, which describes CO-SSL in general.
>
> > What is the motivation behind the RF-ResNet? The paper says it is used for analysis purposes (line 211). But it is also mentioned as one of the contributions on line 64. What is the advantage of RF-ResNet compared to standard ResNet? From what I understood, RF-ResNets are less robust than the standard ResNet and do not bring significant performance improvement. Table 1 is the only place the authors show that RF-ResNet50 is slightly better than ResNet50. However, RF-ResNet 50 also has slightly more parameters than ResNet50 (23.7M vs 23.5M on line 238).
>
> We removed the statement about using the architecture for analysis purposes and clarified in the Introduction that this is part of our contributions. First, RF-ResNet50 outperforms its ResNet50 counterpart on ImageNet-1k (100 epochs), ImageNet-1k (300 epochs) and ImageNet-100 (CO-BYOL) by 0.3% (Table 1), 1% (Table 7) and 1.3% (Table 5), respectively. Table 5 clearly shows that the difference does not come from the number of parameters as RF99-ResNet50v0 uses only 21.38M of parameters (clarified in Section 4.5), without reducing the accuracy on ImageNet-100. Thus, the improvement is significant. A limitation is indeed that this architecture lacks robustness (Section 4.2). Second, because of its design with small receptive field sizes, using a RF-ResNet (with small RFs) demonstrates that CO-SSL infers spatial co-occurrences to build representations (Section 4.4).

---

> > ### Author Response · Authors · 2024-11-26
> > **Rebuttal [2/2]**
> >
> > > How is the generalization of the learned representations from the proposed method to other downstream tasks beyond ImageNet classification ? [...] When an image describes a complicated scene, I'm not sure if it is really beneficial to force all local features to be close to the global feature because there could be more than one object/topic.
> >
> > We added results of a linear probe trained on Places365 in Appendix A.2, which is a widespread dataset for evaluating scene recognition. We notably find that CO-BYOL (RF99-R50) outperforms BYOL by ~1%, suggesting that CO-SSL also works with other semantic classification tasks. Regarding tasks like object segmentation or localization, we did not try but we agree with the reviewer that extending CO-SSL, as discussed below, may be important.
> >
> > > Perhaps it makes more sense to have more than one global feature or set the number of global features as a hyper-parameter. [...] Does it make sense to also use the loss from Equation 2 at intermediate layers?
> >
> > Thank you for these two suggestions which are in line with our thinking. We agree that it would be interesting to explore CO-SSL with different numbers of "global" representations in different intermediate layers of a network. This may allow exploiting spatial regularities at different levels of granularity. For instance, edges may be interesting because they are frequently occurring arrangements of neighboring pixels, and scenes are partly defined by the co-occurrence of entire objects (the co-occurrence of a fridge, a dish washer and a sink provides strong evidence for a kitchen). This is an exciting research direction, but we had to leave it for future work.
> >
> > > The author did not provide any implementation details in terms of how to combine CO-SSL with DINO.
> >
> > Given the genericity of CO-SSL modifications, we added a general explanation of CO-SSL in Section 3.1. We will also release our code based on solo-learn.
> >
> > > What does the performance look like if the local features are also from the same projectors and predictors as the global features?
> >
> > Because of computational constraints, we tried only on ImageNet-100. It leads to a marginal decrease of accuracy (-0.1%); we added the results in Table 6). We lacked time, but we will run experiments on ImageNet-1k as soon as possible.
> >
> > > Why 99 and 29 are chosen as the RF for ResNet 50 and ResNet 18 in the experiments? Does the same conclusion hold for other RF sizes?
> >
> > We can not try all RF sizes in all experiments. We chose the best RF size according to our analysis in Figure 3, A. In Tiny-ImageNet, we chose a RF29-ResNet18 because the image ratio corresponding to its RF size in Tiny-ImageNet (RF=29, 64x64 images, ratio of $(\frac{29}{64}^2=0.2$) is approximately the same as the one of RF99-R50 in ImageNet (RF=99, 224x224 images, $(\frac{99}{224}^2=0.2$). We clarified that in Section 3.3.
> >
> > > What is the depth of the additional MLP added after the pooling layer? What is the effect of the depth of this MLP to the final performance?
> >
> > In Section 3.2, we explain that the structure is similar to one block of convolutions. Thus, this is 3 for a ResNet-50 (like a Bottleneck block) and 2 for a ResNet-18 (Basic Block). It has a marginal importance for CO-BYOL, as removing it (RF99-ResNet50v0) does not significantly change the ImageNet-100 accuracy (Table 6) with respect to RF99-ResNet50.

---

> > > ### Comment · Reviewer_mcd1 · 2024-12-01
> > >
> > > I thank the authors for their rebuttal. I appreciate the improved clarity of the paper and the efforts in answering the questions. However, I'm still concerned about the contribution of RF-ResNet. First, a 0.3%~1.3% improvement over ResNet50 might not be significant enough as the gap is small and the experiments are only conducted once. Second, it lacks robustness. Third, the idea of using small receptive field sizes is not evaluated on the transformer architecture, which is arguably the most popular model nowadays for SSL. Given those points, I'm not sure how significant this contribution is overall speaking.

---

> > > > ### Author Response · Authors · 2024-12-03
> > > >
> > > > We thank the reviewer for their answer.
> > > >
> > > > > However, I'm still concerned about the contribution of RF-ResNet. First, a 0.3%~1.3% improvement over ResNet50 might not be significant enough as the gap is small and the experiments are only conducted once.
> > > >
> > > > We want to clarify that the main contribution of this paper is the CO-SSL objective (cf. introduction). The improvement of CO-BYOL (RF-ResNet) with respect to the main comparison baseline (BYOL) ranges in [1.4-7]% (Table 2), which are large boosts.
> > > >
> > > > RF-ResNet is a second contribution that allows to fully harness CO-SSL for object recognition (cf. Introduction). We agree that 0.3% is relatively small (I-1k, 100 epochs). However, 1% on I-1K (300 epochs) and 1.3% on I-100 are largely recognized as significant in the SSL literature, even with one run. We presume that CO-SSL fully takes advantage of RF-ResNet when training for relatively more epochs and we'll clarify that in Section 4.5.
> > > >
> > > > > Second, it lacks robustness.
> > > >
> > > > True. But our main contribution is CO-SSL. With a ResNet50, CO-SSL shows a robustness improvement of ~10% on average for noise corruptions, small adversarial attacks and internal masking (Table 3, 4).
> > > >
> > > > > Third, the idea of using small receptive field sizes is not evaluated on the transformer architecture, which is arguably the most popular model nowadays for SSL
> > > >
> > > > We agree that it would be interested to combine CO-SSL with transformers that have small receptive fields. We did experiments with standard ViT architectures (cf. our answer to mnvG), which suggest that CO-SSL can boost visual learning with ViTs as well. However, we simply ran out of time to propose/train ViTs with small RFs. We'll keep working in this direction.

---

### Official Review · Reviewer_DxvF · 2024-11-03

**Soundness:** 2
**Presentation:** 2
**Contribution:** 1
**Rating:** 3
**Confidence:** 3

**Summary:**

This paper proposes a novel contrastive learning approach for encouraging spatial cooccurrences by promoting the similarity between local features and global features. To study the impact of local representations, this work introduces RF-ResNet to limit the receptive field of local representations. The proposed framework, CO-SSL, demonstrates superior performance in classification, transfer learning, and robustness. The analysis indicates that RF size is inverse  to the minimum crop ratio.

**Strengths:**

1. This work proposes a novel contrastive learning approach, termed CO-SSL, to encourage spatial cooccurrences. The proposed method demonstrates efficiency in pretraining on ImageNet.
2. It is proved that CO-SSL learns stronger local similarities,  regardless of the neural architecture.

**Weaknesses:**

**The original contribution comparing to BYOL with multi-crops is limited:**
 The loss function between local and global embeddings can be treated as BYOL introducing local crops.

**Conclusion on RF and Crop Size Relationship:**
The conclusion drawn about the relationship between receptive field size and crop size is currently not well-supported. Figure 3(a) shows a general trend but does not clearly demonstrate an inverse correlation. The authors need to strengthen their argument by providing a more rigorous analysis or additional data points that highlight this relationship. Moreover, basing the conclusion solely on ImageNet-100 is limiting, as different datasets may have varying object characteristics that affect the optimal crop size. The authors should consider using additional datasets to validate their findings and discuss how the crop size might be influenced by the specific objects and their arrangements within the dataset images.

**The experiments are not clearly explained :**
1. what the numbers in Figure 2 present is not clear.
2. the calculation of the RF size is missing.

**Questions:**

**Clarity of RF-ResNet Explanation and Figure 2:**
The paper introduces the RF-ResNet architecture, which is designed to limit the receptive field (RF) of local representations. However, the explanation regarding how the RF fields are calculated and what the numbers in Figure 2 represent is indeed unclear. The authors should provide a more detailed explanation of the RF calculation, possibly with a formula or a clear description of the process. Additionally, Figure 2 should be annotated to explain the significance of the numbers, such as the receptive field sizes and how they correspond to the architectural modifications. This will help readers understand the relationship between the architecture and the receptive field size.

**Difference Between RF99-ResNet50 for img224 and ResNet50 for img99:**
The paper compares the performance of RF99-ResNet50 with that of a standard ResNet50. It would be beneficial for the authors to clearly articulate the differences between these two models, especially in the context of input image sizes. Specifically, the RF99-ResNet50 is designed to have a receptive field of 99x99 pixels, while the ResNet50 is not constrained in this way. The authors should explain how these differences impact the models' ability to capture local and global features. The proposed model processes multiple local representations in parallel, similar to dealing with multiple local crops. The authors should provide a direct comparison of the performance of their model with BYOL that uses multi-crops. This comparison should include not only accuracy metrics but also an analysis of sample efficiency and computational costs. If the proposed method offers improvements over BYOL with multi-crops, this should be clearly highlighted and justified with data.

---

> ### Author Response · Authors · 2024-11-26
> **Rebuttal**
>
> We thank the reviewer for their careful reading and suggestions for improving the paper. We discuss each point below.
>
> > The original contribution comparing to BYOL with multi-crops is limited
>
> SSL-mc and CO-SSL both share the idea of better modeling spatial co-occurrences. However, CO-SSL presents two main advantages: 1) CO-SSL is much more sample-efficient (see our answer below); 2) CO-SSL (R50) is less sensitive than SSL-mc to noise corruptions and adversarial attacks (cf. Section 4.2 and Appendix A.7). This is probably because CO-SSL models spatial co-corrences internally, such that their local representations are more similar (Section 4.4 and Fig 3.B). Note that one could also use CO-SSL with multi-crop, but we leave that to future works.
>
> > The conclusion drawn about the relationship between receptive field size and crop size is currently not well-supported. The authors need to strengthen their argument by providing a more rigorous analysis or additional data points that highlight this relationship.
>
> We added points for minimal crop size = 0.3 and removed our statement that there is an “inverse correlation”. Not seeing a strong correlation is intuitively normal as Crop/Resize (C/R) augmentations also have an important regularization effect [1].
>
> However, in Figure 3 and Appendix A.1, we clearly see that CO-BYOL is more robust for large minimum crop sizes, where CO-BYOL outperforms BYOL by up to 12%. This suggests that CO-BYOL with small receptive fields can replace C/R, to some extent.
>
> > Basing the conclusion solely on ImageNet-100 is limiting, [...] The authors should consider using additional datasets to validate their findings and discuss how the crop size might be influenced by the specific objects and their arrangements within the dataset images.
>
> The main take-away of this analysis is that CO-SSL better models spatial co-occurrences than C/R (see above), not to find a general optimal crop size. Image statistics can drastically vary across datasets, and one should ideally always hyper-parameterize the minimum crop size, independently from the model or dataset.
>
> That being said, ImageNet-100 has standard image statistics (as a subset of ImageNet-1k) and we demonstrate that the best minimal crop and RF sizes in ImageNet-100 give strong results in 3 other settings (Table 2). Thus, we think it is unreasonable to spend between 5,000 and 20,000 GPU hours (depending on the dataset) to replicate this study on another dataset.
>
> > What the numbers in Figure 2 present is not clear.
>
> We updated the Figure and the captions to clarify all symbols.
>
> > The calculation of the RF size is missing.
>
> For completeness, we added the used formula in Appendix B.1 (from [2]).
>
> > Additionally, Figure 2 should be annotated to explain the significance of the numbers, such as the receptive field sizes and how they correspond to the architectural modifications. This will help readers understand the relationship between the architecture and the receptive field size.
>
> To keep Figure 2 readable, we decided to add text. In Section 3.2,  we now explain that we have to decrease convolution strides and kernel sizes in convolution layers to decrease the final receptive field sizes. We provide the intuition in Appendix B.1: that is because, at each layer, RFs become larger by the product of all previous strides and the layer kernel size (compared to RFs of the previous layer). For a more thorough explanation, we refer to [2].
>
> > Specifically, the RF99-ResNet50 is designed to have a receptive field of 99x99 pixels, while the ResNet50 is not constrained in this way. The authors should explain how these differences impact the models' ability to capture local and global features.
>
> In Section 4.4, we thoroughly analyze the differences between ResNet50 and RF99-ResNet50 with CO-BYOL. To sum up, RF99-ResNet50 is forced by design to capture local features while ResNet50 naturally captures slightly larger local features. Thus, in both cases but especially for RF99-ResNet50, the models must infer the spatial regularities of features to minimize the loss function. This offers an explanation of why CO-BYOL (RF99-50) performs better than CO-BYOL (R50) in Figure 3, Table 1, 6 and 7.
>
> > The authors should provide a direct comparison of the performance of their model with BYOL that uses multi-crops[...]  with  sample efficiency and computational costs.
>
> We added a rigorous comparison in Appendix A.3, in terms of memory, FLOPs, sample efficiency and accuracy. Briefly, CO-BYOL (RF99-R50) requires more FLOPs, but is more performant and memory-efficient. CO-BYOL (R50) presents benefits over BYOL-mc in all aspects.
>
> [1] Hernández-García, A., & König, P. (2018). Data augmentation instead of explicit regularization. arXiv preprint arXiv:1806.03852.
>
> [2] Araujo, A., Norris, W., & Sim, J. (2019). Computing receptive fields of convolutional neural networks. Distill, 4(11), e21.

---

> > ### Comment · Reviewer_DxvF · 2024-12-02
> >
> > Thank you for response. Although the revision has improved, the contribution is still limited. The authors should answer the main contribution clearly. If the main contribution is the objective (CO-BYOL), the models for comparison should be optimal (trained on IN1K and long enough). If the main contribution is the efficiency, the reduced time varing model size should be demonstrated. A deeper study is needed to reach the acceptance line.

---

> > > ### Author Response · Authors · 2024-12-02
> > >
> > > We thank the reviewer for their response.
> > >
> > > >  The authors should answer the main contribution clearly. If the main contribution is the objective (CO-BYOL), the models for comparison should be optimal (trained on IN1K and long enough). If the main contribution is the efficiency, the reduced time varing model size should be demonstrated. A deeper study is needed to reach the acceptance line.
> > >
> > > We are interested in sample-efficient and robust visual learning. We are open to make that clearer in the introduction. CO-BYOL is largely better (~1.5%) than BYOL for IN1K with 100 and 300 epochs pre-training (Table 1 & 7) and in three other settings (Table 2). That being said, a fair comparison to BYOL-mc would require a CO-BYOL-mc. CO-SSL (R50) is also systematically more robust to noise corruptions, adversarial attacks and internal masking than BYOL and BYOL-mc (by 10-15% on average).

---

### Official Review · Reviewer_mnvG · 2024-11-04

**Soundness:** 3
**Presentation:** 3
**Contribution:** 3
**Rating:** 6
**Confidence:** 4

**Summary:**

This paper leverages the spatial co-occurrences of SSL by focusing on local representations. It proposes a new method to learn the correspondence between local features to global features within an image, based on a modified ResNet that bounds local receptive field. The paper demonstrates the performance of the proposed CO-SSL, and its robustness against corruptions.

**Strengths:**

1. The analytical experiments (robustness, similarities, saliencies) are insightful. From visualizations in Figure 4, CO-SSL works well.

2. The work has promising extensions, and probably can shed light on modifying ViTs to enhance local focus and processing.

3. This paper is trying to make solid contributions as it not only proposes new objectives but also a new architecture, thanks for the hard work.

**Weaknesses:**

1. Referring to Table 2, why applying CO-SSL to DINO does not improve the baseline method as effectively as that for BYOL? This also makes me curious about the effectiveness of CO-SSL on other similar work with two models, e.g., Barlow Twins. I expect CO-SSL to generalize to other SSL methods as well, or the contribution is narrowed.

2. Following (1), while the authors claim that "in principle, the approach is adaptable to most SSL methods,", we don't know the effectiveness and performance. In later experiments, the authors mainly focuses on improving BYOL. Can the authors provides results with DINO+CO-DINO, and more? While I in general appreciate the proposed objectives and modified architectures, given the insufficient experiment settings, I am skeptical whether the proposed methods is limited to BYOL and cannot generalize to other SSL methods.

3. It would be great if the authors can analyze and explain the co-occurrences more, specifically, is it brought by convolutions? Will the observations and solutions here hold for transformers?

**Questions:**

While I acknowledge the hard work and the promising future extensions of this works, there are several concerns that need to be addressed, majorly regarding its generalizability to many SSL methods. This paper has a narrowed demonstration of applying CO-SSL to only one or two methods, which cannot convince the audience about its capability effectively. Please see the weakness section for more details.

---

> ### Author Response · Authors · 2024-11-26
> **Rebuttal**
>
> We thank the reviewer for the responses and suggestions for improving the paper. We discuss each point below.
>
> > Why applying CO-SSL to DINO does not improve the baseline method as effectively as that for BYOL?
>
> DINO has many more hyper-parameters than common SSL methods (like BYOL, SimCLR, VicReg, Barlow Twins…): absence/presence of batch normalization in projection heads, bottleneck layer size, absence/presence of normalization in last layer of the projection head, temperature, temperature scheduler, center momentum coefficient etc... We suspect that the default DINO was designed to work with Crop/Resize and we could not explore all these choices with CO-SSL.
>
> > This also makes me curious about the effectiveness of CO-SSL on other similar work with two models.
>
> Following your suggestion, we added results of MoCoV3 [1] and CO-MoCoV3 in Appendix A.1. We finished experiments on Tiny-ImageNet , ImageNet-100 and ImageNet-1k (100%) for MoCoV3 and CO-MoCoV3 and we will add missing experiments in the next version of the paper. CO-MoCoV3 consistently outperforms MoCoV3, confirming our previous results.
>
> We also started experiments comparing CO-VICReg to VICReg (a SSL method similar to Barlow Twins) on Tiny-ImageNet. We observe that CO-VICReg improves over VICReg by 1.1%. We ran out of time to try out the experiments on datasets with bigger images.
>
> Overall, this confirms our conclusions.
>
> >  Following (1), while the authors claim that "in principle, the approach is adaptable to most SSL methods,", We don't know the effectiveness and performance. In later experiments, the authors mainly focuses on improving BYOL. Can the authors provides [> Section 4.1] results with DINO+CO-DINO ?
>
> In practice, CO-BYOL is the main instance model of CO-SSL and we can not replicate all analysis for several models for computational reasons. However, we added robustness experiments with DINO and CO-DINO in Appendix A.7. In short, we obtain similar results as for BYOL and CO-BYOL. There is one exception, which is the robustness to internal masking where CO-DINO performs poorly (compared to DINO and CO-BYOL). The reason is currently unclear to us.
>
> > It would be great if the authors can analyze and explain the co-occurrences more, specifically, is it brought by convolutions?
>
> In Section 4.3, we show that limiting the size of the theoretical receptive field of local representations is crucial to fully leverage CO-BYOL. In Section 4.4, we also find that local representations reflect relatively local features rather than global features in a ResNet50. Thus, CO-BYOL can also use local representations from a ResNet50 to model spatial co-occurrences. We presume that this locality bias roots in the convolution layers.
>
> > Will the observations and solutions here hold for transformers?
>
> In preliminary experiments, we observed that the effective receptive fields (ERFs) of the deepest patch representations (except CLS) of a ViT-S/16 (ImageNet-1k) mostly highlights the 16x16 pixels of the given patch location and, to a smaller extent, the center of the image that contains the main semantic content. This is different from the Gaussian ERFs in local representations of ResNets (Fig 4.) and is probably due to the use of long-range attention layers (cf. also [2]).  It is unclear to us whether and how CO-SSL can use representations with such ERFs. We suspect that using Swin transformers [3], which have a hierarchical design with short-range attention, may be important to fully leverage CO-SSL. We agree this is an interesting direction for future work.
>
> [1] Chen, X., Xie, S., & He, K. (2021). An empirical study of training self-supervised vision transformers. In Proceedings of the IEEE/CVF international conference on computer vision (pp. 9640-9649).
>
> [2] Raghu, M., Unterthiner, T., Kornblith, S., Zhang, C., & Dosovitskiy, A. (2021). Do vision transformers see like convolutional neural networks?. Advances in neural information processing systems, 34, 12116-12128.
>
> [3] Liu, Z., Lin, Y., Cao, Y., Hu, H., Wei, Y., Zhang, Z., ... & Guo, B. (2021). Swin transformer: Hierarchical vision transformer using shifted windows. In Proceedings of the IEEE/CVF international conference on computer vision (pp. 10012-10022).

---

> > ### Comment · Reviewer_mnvG · 2024-11-26
> >
> > I thank the authors for their clarifications and additional results. I assume for added MoCo-v3 results, the authors train with ResNet50? If the authors train with ViTs, please specify how the architecture is modified to incorporate RF designs as it can be a valuable contribution. And could you please compare with MoCo-v3-mc setting? I also look forward to new results on the mentioned CO-VICReg (vs vanilla and mc) as the deadline has been extended too.
> >
> > In short, the added Table 8, 12, and 13 address my concerns and I tend to believe that CO-SSL can apply to major SSL methods; it is slightly better than mc strategy on linear evaluation and is notably better than mc regarding robustness. I want to raise my rating a little bit; I cannot raise more because (a) I expect more empirical results comparing different SSLs vs. SSLs+mc vs. CO-SSLs (so more settings in Table 1, but I understand the time constraints), and (b) I expect the authors digging deeper into formulating global and local features so that one can derive RF variants for different architectures equally readily.

---

> > > ### Author Response · Authors · 2024-12-03
> > >
> > > We used ResNet50 for MoCo-v3 (ResNet18 in Tiny-ImageNet).  We kept experimenting with MoCo-V3 and we have so far (top 1 linear accuracy):
> > >
> > > |        | MoCoV3 | MoCoV3-mc | CO-MoCoV3 |
> > > | ------  | ------ | ------ | ------ |
> > > | Tiny-I | 51.78 | 57.87 | 56.03 |
> > > | I-100 | 83.36 | 83.22 | 87.34 |
> > > |I-1K (100%) | 69.77 | 70.4 | 70.89
> > >
> > > We plan to extend the comparison of MoCo-V3 and VICReg as we did for other methods as soon as possible. Training a standard ViT-small with CO-BYOL reaches 69.54% of accuracy versus 68.97% for BYOL. For Swin-T, CO-BYOL also outperforms BYOL by 0.4% (67.53% versus 67.09%).
> > >
> > > We thank the reviewer for their suggestions and raising their score. We will keep working in these directions.

---

### Official Review · Reviewer_UjWt · 2024-11-04

**Soundness:** 3
**Presentation:** 2
**Contribution:** 2
**Rating:** 5
**Confidence:** 3

**Summary:**

The paper presents CO-SSL, an SSL method that learns similar representations for frequently co-occurring visual features, enhancing robustness and outperforming previous models (71.5% Top-1 accuracy on ImageNet-1K). It introduces RF-ResNet to control the receptive field size and analyze local representation impact. Results indicate CO-SSL’s high redundancy in local features contributes to robustness, suggesting co-occurrence learning as a powerful unsupervised category learning principle.

**Strengths:**

1. the idea of co-occurrence is important in SSL
2. the title is clear and intuitive

**Weaknesses:**

1. The spatial co-occurrence has been identified as useful before. E.g. in SimCLR cropping has shown it importance. [1] further proposes SSL training on patches. The proposed method should compare with such methods and include a better analysis of why the proposed multiple head is better, in addition to short empirical study in Fig3.
2. The experimental improvement in Table 1 is also limited, which is a) related to point 1; b) potentially indicates that the original design (e.g. BYOL, simclr) implicitly considers spatial occurrence with patch/cropping. The design may not be very necessary, or it is even a duplicate.
3. Please include ablation study on number of heads (e.g. Fig.1 only have 2), conv layers to have different projection and number of projection embeddings. See [2] for an example showing different performance at conv3 conv4

minors:
1. Please include caption below figure and not use see text for details and omit captions.

summary:
While the topic is of importance, the spatial co-occurrence in SSL is not new and explored already. The proposed method seems to be a special kind but at current format analysis such as the best design is not well explored.

[1] Bag of Image Patch Embedding Behind the Success of Self-Supervised Learning

[2] Pose-Aware Self-Supervised Learning with Viewpoint Trajectory Regularization, ECCV 2024

**Questions:**

see Weaknesses.

---

> ### Author Response · Authors · 2024-11-26
> **Rebuttal**
>
> We thank the reviewer for the responses and suggestions for improving the paper. We discuss each points below.
>
> > The spatial co-occurrence has been identified as useful before. E.g. in SimCLR cropping has shown it importance. [1] further proposes SSL training on patches. The proposed method should compare with such methods and include a better analysis of why the proposed multiple head is better, in addition to short empirical study in Fig3. The experimental improvement in Table 1 is also limited, which is a) related to point 1; b) potentially indicates that the original design (e.g. BYOL, simclr) implicitly considers spatial occurrence with patch/cropping. The design may not be very necessary, or it is even a duplicate.
>
> We argue in the introduction that BYOL, SimCLR etc… model spatial co-occurrences with Crop/Resize. But CO-SSL does it better. We already compared CO-SSL to Crop/Resize (x2) in all our experiments of Sections 4.1, Section 4.2, Appendix A.1 and Appendix A.7: CO-BYOL, CO-DINO, CO-MoCoV3 consistently outperform their SSL counterparts. For instance, CO-BYOL improves over BYOL with Crop/Resize (x2) by [1.4-7.3]% on the four considered datasets (Table 2). This is a significant gain. The gain is smaller with respect to MEC or Matrix-SSL in Table 1. However, their contribution is orthogonal to ours and we leave extending their approaches with CO-SSL for future work.
>
> BagSSL [1] proposes to model spatial co-occurrences by applying SSL on local crops of the same size within an image; CO-BYOL outperforms their patch-based BYOL pre-training by ~10% on Imagenet-100 (Table 2 versus their results in [1]).
>
> > Include a better analysis of why the proposed multiple head is better, in addition to short empirical study in Fig3.
>
> Our analysis exhibits two reasons for our result: 1) Crop/resize (x2) and [1] are limited to one spatial co-occurrence per image (two crops). In contrast, Table 5 (n^2) highlights that CO-BYOL can benefit from more co-occurrences of visual features per image. We emphasized this aspect in our revision by moving key results into Section 4.3; 2) It creates more redundant local representations (cf. Section 4.4).
>
> > Please include ablation study on number of heads (e.g. Fig.1 only have 2), conv layers to have different projection and number of projection embeddings.
>
> CO-SSL uses two projection heads and it is unclear to us how and why to increase this number. Could the reviewer clarify their idea? Please, note that we also applied CO-SSL in other layers in Appendix A.6 and we added results when using one head, i.e.  the same head to compute local and global embeddings (Table 6, it leads to -0.1% of accuracy).
>
> > See [2] for an example showing different performance at conv3 conv4.
>
> We added in Appendix A.2 results with an off-line linear probe trained on layer3 and layer4. Briefly, these layers perform worse than the final layer of the visual backbone, which is also consistent with the conclusion of [2] for semantic classification.
>
> > Please include caption below figure and not use see text for details and omit captions.
>
> We added accurate captions to all figures.

---

> > ### Comment · Reviewer_UjWt · 2024-11-26
> >
> > Thank you for the rebuttal. However, as not all of my concerns were addressed, I will maintain my score and lean toward rejection:
> >
> > **Comparison with BagSSL**
> >
> > The authors state: "We argue in the introduction that BYOL, SimCLR, etc., model spatial co-occurrences with Crop/Resize" and "BagSSL [1] proposes to model spatial co-occurrences by applying SSL on local crops of the same size within an image; CO-BYOL outperforms their patch-based BYOL pre-training by ~10%."
> >
> > BagSSL employs a similar approach by modeling spatial co-occurrences.
> >
> > A direct theoretical and methodological comparison with BagSSL is missing, despite the apparent similarity.
> >
> > The empirical comparison, relying solely on reported numbers, is potentially inaccurate due to differing networks and experimental settings. A proper numerical analysis is also lacking.
> >
> > **Ablation Study on Multiple Heads**
> >
> > Since multiple augmentations are possible, the initial review emphasized the relevance of an ablation study on the number of heads.
> >
> > The authors dismissed this without conducting or discussing the analysis, which weakens the argument.

---

> > > ### Author Response · Authors · 2024-11-29
> > >
> > > We thank the reviewer for their quick answer, which allows us to further elaborate to address their concern.
> > >
> > > > BagSSL employs a similar approach by modeling spatial co-occurrences.
> > >
> > > BagSSL is very different from CO-SSL. BagSSL still uses Crop/Resize to model spatial co-occurrences, while CO-SSL uses local representations. Our method is actually closer to Crop/Resize than BagSSL because the only modification (during training) of BagSSL with respect to Crop/Resize is that it models spatial co-occurrences with a fixed scale. In contrast CO-SSL models spatial co-occurrences with multiple scales because 1- CO-SSL is combined with multi-scale Crop/Resize and 2- we make global and local representations similar. We will highlight these differences in Section 2 (Related works).
> > >
> > > > A direct theoretical and methodological comparison with BagSSL is missing, despite the apparent similarity. The empirical comparison, relying solely on reported numbers, is potentially inaccurate due to differing networks and experimental settings. A proper numerical analysis is also lacking.
> > >
> > > ImageNet-100 results can be compared. The original results of BagSSL also use the ResNet50 architecture, with image patches of the same size (100x100)  as the RF of RF99-ResNet50 (99x99) and they pre-train with the same number of epochs (400).
> > >
> > > Our main comparison baseline (“Multiscale” Crop/Resize) also outperforms BagSSL in their paper. The main contribution of BagSSL _is not_ to be better than Crop/Resize at modelling spatial co-occurrences, but to “establish a formal connection between joint-embedding SSL and the co-occurrence of image patches”. We will add a statement in Section 4.1 to clarify that Crop/Resize is the strongest comparison baseline for modeling spatial co-occurrences, to the best of our knowledge.
> > >
> > > In sum, CO-SSL is closer to Crop/Resize than BagSSL, Crop/Resize is stronger than BagSSL and CO-BYOL outperforms BagSSL (BYOL) by a huge gap of ~10% on a similar setting based on reported numbers. Thus, we think it is unreasonable to spend between 10,000 and 30,000 GPU hours (with a fair hyperparameter search) to run BagSSL in all our settings.
> > >
> > > > Since multiple augmentations are possible, the initial review emphasized the relevance of an ablation study on the number of heads. The authors dismissed this without conducting or discussing the analysis, which weakens the argument.
> > >
> > > We already added a comparison between 1 and 2 heads. We do not understand the original comment for more than 2 heads or the connection with “multiple augmentations”. We have explicitly asked the reviewer to explain what they mean. We’d be happy to answer during the discussion period if the reviewer clarifies what they mean and why it is relevant.

---

### Author Response · Authors · 2024-11-26
**Global response**

We thank the reviewers for their feedback on the paper. We find it encouraging that reviewers find that our paper is easy-to-follow (mcd1) and tackles an important topic (UjWt) with a simple method (mcd1). Furthermore, reviewers find our results important (mcd1) and our analytical experiments insightful (mnvG).

There is a general agreement that we can improve the presentation of the method, especially of the captions and figures (UjWt, mcd1, DxvF). In the new version of the paper, we clarified the points raised by the reviewers and overhauled the figures and captions.

In the revised paper, we color in blue our modifications. We will address the rest of your comments in individual answers.

---

### Meta-Review · Area_Chair_HsUX · 2024-12-08

**Metareview:**

This submission introduces CO-SSL, a method in self-supervised learning (SSL) aimed at aligning local and global visual representations to enhance robustness and performance. The paper demonstrates improvements over existing methods, particularly BYOL, on datasets such as ImageNet. It also proposes RF-ResNet, an architecture designed to limit the receptive field size, enabling better alignment with the CO-SSL approach.

Strengths:
* The proposed CO-SSL demonstrates moderate improvements in some SSL tasks, particularly in robustness to corruptions and adversarial attacks.
* The paper includes analytical experiments highlighting the method's potential for spatial co-occurrence modeling.
* Efforts were made to extend the approach to other SSL methods like DINO and MoCoV3.

Weaknesses:
* Generalizability: The method's application beyond convolutional architectures, such as transformers, is not thoroughly explored, raising questions about its broader applicability.
* Presentation: Several reviewers noted issues with the clarity of figures, captions, and the overall presentation, suggesting that the paper could benefit from more detailed explanations and clearer visuals.
* Comparison and Ablation: There is a lack of comprehensive comparison with similar methods like BagSSL, and the ablation studies on crucial aspects such as the number of projection heads are missing or inadequately addressed.
* Robustness and Contribution of RF-ResNet: While CO-SSL shows robustness, the specific contribution of RF-ResNet to this robustness and overall performance is questioned due to its limited improvement and lack of robustness compared to standard ResNet.

In conclusion, while the paper presents an interesting approach to SSL, the lack of clear novelty, inadequate exploration of broader applicability, and presentation issues lead to a recommendation for rejection. The authors would need to address these concerns more comprehensively to make a case for acceptance in future submissions.

**Additional Comments On Reviewer Discussion:**

Points Raised by Reviewers:

UjWt:
Questioned the novelty of CO-SSL compared to existing methods like BagSSL, suggesting a need for direct comparison.
Requested an ablation study on the number of heads.

mnvG:
Sought clarity on why CO-SSL didn't improve DINO as significantly as BYOL.
Requested more experiments with different SSL methods to show generalizability.

DxvF:
Doubting the contribution of RF-ResNet, especially concerning robustness.
Requested clearer explanations of figures and receptive field calculations.

mcd1:
Criticized the presentation, asking for improved figures and clearer contribution statements.
Questioned the significance of RF-ResNet's contribution and its robustness.

Authors' Responses:

UjWt:
Clarified that CO-SSL differs from BagSSL by using local representations at multiple scales, not just fixed-size patches.
Added comparisons but argued against extensive new experiments due to resource constraints.

mnvG:
Provided more experiments with MoCoV3 and CO-MoCoV3, showing improvement, though not all experiments were completed due to time constraints.
Discussed potential for transformers but indicated further research needed.

DxvF:
Clarified RF-ResNet's role and provided more details on architecture modifications.
Added comparisons with multi-crop BYOL, focusing on efficiency and robustness.

mcd1:
Improved figures and captions for clarity.
Clarified the general applicability of CO-SSL and the specific contributions of RF-ResNet.
Added results for scene recognition to address generalizability concerns.

---

### Decision · Program_Chairs · 2025-01-22

Reject